# Diverse Populations of *Staphylococcus pseudintermedius* Colonize the Skin of Healthy Dogs

Norma Fàbregas,[a] Daniel Pérez,[b] Joaquim Viñes,[a]* Anna Cuscó,[a]§ Lourdes Migura-García,[c,d] Lluís Ferrer,[b] Olga Francino[e]

[a]Vetgenomics, Edifici EUREKA, PRUAB, Universitat Autònoma de Barcelona (UAB), Bellaterra, Barcelona, Spain
[b]Department of Animal Medicine and Surgery, Universitat Autònoma de Barcelona (UAB), Bellaterra, Barcelona, Spain
[c]Joint Research Unit IRTA-UAB in Animal Health, Animal Health Research Centre (CReSA), Universitat Autònoma de Barcelona (UAB), Bellaterra, Barcelona, Spain
[d]IRTA, Animal Health Program, Animal Health Research Centre (CReSA), Universitat Autònoma de Barcelona (UAB), Bellaterra, Barcelona, Spain
[e]SVGM, Molecular Genetics Veterinary Service, Universitat Autònoma de Barcelona (UAB), Bellaterra, Barcelona, Spain

**ABSTRACT** *Staphylococcus pseudintermedius* is a commensal bacterium of the canine skin but is also a key opportunistic pathogen that is responsible for most cases of pyoderma in dogs. The current paradigm indicates that infection arises when predisposing factors alter the healthy skin barrier. Despite their importance, the characteristics of the *S. pseudintermedius* populations colonizing the skin of healthy dogs are yet largely unknown. Here, we retrieved 67 complete circular genomes and 19 associated plasmids from *S. pseudintermedius* isolated from the skin of 9 healthy dogs via long-reads Nanopore sequencing. Within the *S. pseudintermedius* populations isolated from healthy skin, multilocus sequence typing (MLST) detected 10 different STs, distributed mainly by the host. 39% of the 18 representative genomes isolated herein were methicillin-resistant *S. pseudintermedius* (MRSP), and they showed, on average, a higher number of antibiotic resistance genes and prophages than did the methicillin-sensitive (MSSP). In summary, our results revealed that the *S. pseudintermedius* populations inhabiting the skin of healthy dogs are relatively diverse and heterogeneous in terms of MLST and methicillin resistance. In this study, all of the 67 commensal *S. pseudintermedius* populations that were isolated from healthy dogs contained antibiotic resistance genes, indicating the extent and severity of the problem of antimicrobial resistance in staphylococci with zoonotic potential.

**IMPORTANCE** *Staphylococcus pseudintermedius* is a commensal canine bacterium that can become an opportunistic pathogen and is responsible for most cases of canine pyoderma. It can also cause occasional zoonotic infections. Infections caused by antibiotic-resistant *Staphylococcus* are a global concern. Skin commensal *Staphylococcus pseudintermedius* is understudied. To provide insight into the commensal strains circulating in healthy dogs, we performed whole-genome sequencing of 67 *S. pseudintermedius* isolates from different skin sites in 9 healthy dogs. Through the bioinformatic analysis of these genomes, we identified a genomic diversity that is more complete than those afforded by traditional molecular typing strategies. We identified 7 new STs. All of the isolates harbored genes associated with antibiotic resistance, and 39% of the representative genomes were methicillin-resistant. Our data provide critical insights for future skin infection control and antibiotic surveillance within veterinary medicine.

**KEYWORDS** *Staphylococcus pseudintermedius*, pyoderma, skin, WGS, long-reads, MLST, MRSP

The incidence of bacterial resistance to antibiotics is increasing in both human and veterinary medicine. In fact, the World Health Organization has defined antibiotic resistance as one of the greatest threats to global health (https://www.who.int/news-room/fact-sheets/detail/antibiotic-resistance). Thus, the reduced and rational use of antibiotics in

Address correspondence to Norma Fàbregas, norma.fabregas@vetgenomics.com, or Olga Francino, Olga.Francino@uab.cat.

*Present address: Joaquim Viñes, Servei de Microbiologia-CDB, Hospital Clínic de Barcelona, Barcelona, Spain, or ISGlobal, Institut for Global Health, Barcelona, Spain.

§Present address: Anna Cuscó, Institute of Science and Technology for Brain-Inspired Intelligence, Fudan University, Shanghai, People's Republic of China.

The authors declare no conflict of interest.

veterinary medicine is one of the main strategies by which to reduce bacterial resistance. One of the main indications for the use of antibiotics in veterinary medicine is a bacterial skin infection, or pyoderma, caused mainly by *Staphylococcus pseudintermedius*. This commensal bacterium that inhabits the skin of dogs is also a key opportunistic pathogen (1). The current paradigm indicates that infection arises when predisposing factors, such as atopic dermatitis, surgical procedures, or immunosuppressive disorders, alter the skin barrier (2). The increasing emergence of methicillin-resistant *S. pseudintermedius* (MRSP) is becoming a severe challenge to sustained canine health in veterinary medicine (3–5), and it highlights the needs for accurate long-term surveillance (6) and for the use of alternatives with which to treat such infections (7). Understanding the progression process from colonization to infection can guide clinicians and microbiologists to develop new strategies by which to control canine pyoderma, aiming to increase treatment effectiveness and reduce the development of antibiotic resistance. Furthermore, *S. pseudintermedius* is a zoonotic pathogen and a key candidate for One Health approaches (8).

The first *S. pseudintermedius* complete genome report described the ED99 strain, with a genome size of 2.572 Mbp, including insertion and mobile genetic elements, transposons mediating resistance to antibiotics, a family of reverse transcriptases, and a putative integrated plasmid (9). *S. pseudintermedius* ED99 encoded several predicted toxins, exoenzymes, and cell wall-associated proteins (9). Moreover, it is well-described that *S. pseudintermedius* has the ability to form biofilms (10), which hampers the success of antibiotic treatments. To date, numerous studies have reported genomic information regarding *S. pseudintermedius* isolates from the skin of dogs with pyoderma, contributing to a better understanding of the virulence factors and epidemiological distribution of the different pathogenic strains (1, 3, 5, 11–15).

Multilocus sequence typing (MLST) is a powerful DNA sequence-based technique for analyzing bacterial populations and epidemiological studies (16). MLST is frequently used to analyze the clonal associations between strains of clinically relevant microbial species (16). The first specific MLST database for *S. pseudintermedius* started in 2013 (17), and it contains 2,621 MLST profiles as of January 2023 (https://pubmlst.org/organisms/staphylococcus-pseudintermedius). The most prevalent *S. pseudintermedius* MRSP clones that were isolated from cases of pyoderma are ST68 in North America, ST45 in Asia, and ST71 in Europe and Oceania (4, 6, 18, 19). Additionally, ST258 and ST551 are also emerging in Europe (15, 20–24).

Numerous scientific studies have reported whole-genome sequencing (WGS) and MLST data, mainly from MRSP pathogenic genotypes and strains isolated from dogs with skin infections, and these provide an essential framework for investigations into the molecular pathogenesis of canine bacterial pyoderma. However, WGS studies on *S. pseudintermedius* commensal populations that naturally inhabit the skin of healthy dogs remain limited. The current hypothesis stands that pyoderma is initiated from commensal *S. pseudintermedius* that evolve into pathogenic strains. To test this hypothesis, we need to better understand which and how bacterial populations live on the skin of healthy dogs before the infection arises.

While most of the WGS studies on *S. pseudintermedius* that have been reported to date are based on short-read sequencing technologies, long-read sequencing Oxford Nanopore Technology (Nanopore) is more affordable and allows for the sequencing of single native DNA molecules through polymerase chain reaction (PCR)-free protocols (25). The main concern in Nanopore sequencing is the error rate, which is higher than that of Illumina short-read sequencing. Thus, short-read polishing is needed to correct insertion and deletion errors that are derived from homopolymer regions (25). However, Nanopore has recently improved its chemistry and analysis software, which currently allows for nearly-finished bacterial genomes without short-read polishing (26, 27) as well as for a better assembly of the associated plasmids and virus sequences (https://github.com/fenderglass/Flye).

In this context, this study aimed to characterize populations of *S. pseudintermedius* that were obtained from different anatomical sites of healthy dogs. Thus, we carried out a comprehensive genomic and functional data analysis of high-quality and complete genomes

from *S. pseudintermedius* that were isolated from the skin of healthy dogs using Nanopore long-reads sequencing only. Additionally, pangenome and functional enrichment analyses were performed to shed new light on the pathogenic mechanisms of *S. pseudintermedius*.

## RESULTS

**Complete *de novo* assembled genomes and associated plasmids of *Staphylococcus pseudintermedius* isolated from the skin of nine healthy dogs.** To characterize the skin commensal *S. pseudintermedius* genome, we sampled different skin sites from 9 healthy dogs that belonged to different breeds, and we isolated 67 colonies of *S. pseudintermedius*. After the phenotypical identification of *S. pseudintermedius* colonies in blood agar culture medium and DNA extraction, DNA libraries were prepared and sequenced with Nanopore. The average read $N_{50}$ was 5,270.77 bp, with 133,226.86 reads per sample. A total of 67 genomes of *S. pseudintermedius* were *de novo* assembled with Flye 2.8 (12) and were reassembled herein with Flye 2.9. New assembly median values were 156× coverage, 2.60 Mbp genome size, 37.60% guanine-cytosine (GC) content, 99.43% completeness (96.95 to 99.43%), 0% contamination (0 to 1.14%), 2,401 coding sequences, 19 complete rRNAs, 59 tRNAs, and 4 ncRNAs (Table S1). The sequencing coverage exceeded a sequencing depth of 30× in all of the samples. After polishing, all of the corrected genome assemblies showed a completeness above 90% and a contamination below 5% (Table S1), indicating that they were nearly complete genomes with low contamination (28). Thus, they were considered for further analyses. These genomes were further classified as complete genomes via NCBI Prokaryotic Genome Annotation Pipeline (PGAP) (NCBI BioProject: PRJNA685966).

All of the 67 *S. pseudintermedius* genomes were assembled into a main circular and complete contig, corresponding to the bacterial chromosome (Table S1). Flye assemblies showed that 16 out of the 67 isolates contained 20 additional, smaller contigs that had higher coverage than did the main contig, which is usually indicative of high copy number plasmids, except for HSP283 (Table S2). BLASTn confirmed that 19 out of the 20 small contigs aligned with previously reported plasmid sequences. Annotation revealed that 19 out of the 20 small contigs contained hypothetical proteins, 17 contained replication proteins, 6 showed small drug resistance family proteins, and 2 contained the aminoglycoside 6-nucleotidyltransferase *ANT(6)-I* antibiotic resistance gene (Table S2). These small contigs were further classified as plasmids via PGAP annotation (NCBI BioProject: PRJNA685966), except for the HSP283 small contig, which encoded a Retron-type, RNA-directed DNA polymerase that corresponded to chromosomal DNA. By following the criteria used in this study, we concluded that the HSP283 small contig was not a plasmid but rather corresponded to misassembled chromosomal DNA.

Taken together, the Nanopore-only long-reads sequencing allowed for the successful *de novo* assembly of 67 *S. pseudintermedius* complete and circular genomes as well as for the identification of 19 plasmids that were associated with their host genome.

**_S. pseudintermedius_ from healthy dogs show diverse MLSTs that are specific to each dog.** For each genome assembly, the allele numbers and MLSTs were determined, based on the *S. pseudintermedius* PubMLST database. We annotated and reported to PubMLST new combinations of known alleles, resulting in 7 new MLSTs from 6 dogs (ST2175, ST2176, ST2177, ST2178, ST2179, ST2180, and ST2181), whereas 3 of the MLSTs were already described (ST294, ST551, and ST2016). The rest remained as unknown MLSTs because they contained new alleles that were not present in the PubMLST database (Table 1; Table S1).

When analyzing the different MLSTs that were identified in this study, a common MLST could not be identified for all of the isolates. In fact, the MLSTs were mainly distributed by dog. Each MLST was specific to each dog (Table 1), except for one single isolate from Dog 5, which shared the ST2178 with 7 other isolates from Dog 6.

Moreover, the different *S. pseudintermedius* isolates clustered by MLST (Fig. 1). For instance, the ST1026 isolates from Dog 7 clustered together. ST551 from Dog 6 clustered together and apart from ST2178 from the same dog, which in turn clustered together with ST2178 from Dog 5. Furthermore, Dog 8 appeared to show more divergence,

**TABLE 1** Comparison of *S. pseudintermedius* populations genomic information, grouped by MLST variability and host, with the methicillin resistant *S. pseudintermedius* (MRSP) presented in bold[a]

| HOST | n | MLST | Met R genotype | MRSP | MSSP | Median genome size (Mbp) | Median no. of prophages | Median no. of plasmids | Median of ARG | Median of VF |
|------|---|------|----------------|------|------|--------------------------|-------------------------|------------------------|---------------|--------------|
| Dog 1 | **13** | **Unknown** | **MRSP** | **13** | | **2.596** | **0** | **0** | **9** | **43** |
| Dog 2 | 1 | Unknown | MSSP | | 1 | 2.574 | 1 | 0 | 1 | 41 |
| Dog 3 | 1 | Unknown | MSSP | | 1 | 2.471 | 1 | 0 | 1 | 44 |
| | 4 | 2175 | MSSP | | 4 | 2.473 | 1 | 0 | 1 | 44 |
| Dog 4 | 1 | Unknown | MSSP | | 1 | 2.524 | 1 | 0 | 1 | 41 |
| | 1 | 2176 | MSSP | | 1 | 2.612 | 2 | 0 | 2 | 43 |
| Dog 5 | **4** | **2177** | **MRSP** | **4** | | **2.751** | **3** | **0** | **10** | **41** |
| | **1** | **Unknown** | **MRSP** | **1** | | **2.751** | **3** | **0** | **10** | **41** |
| | 1 | 2178 | MSSP | | 1 | 2.604 | 2 | 1 | 2 | 41 |
| Dog 6 | **2** | **Unknown** | **MRSP** | **2** | | **2.647** | **1** | **0** | **9** | **42** |
| | **8** | **551** | **MRSP** | **8** | | **2.874** | **4** | **1** | **10** | **41** |
| | 7 | 2178 | MSSP | | 7 | 2.604 | 2 | 1 | 2 | 41 |
| Dog 7 | **11** | **1026** | **MRSP** | **11** | | **2.599** | **3** | **0** | **3** | **39** |
| Dog 8 | 4 | 2179 | MSSP | | 4 | 2.587 | 1 | 0 | 1 | 40 |
| | 4 | Unknown | MSSP | | 4 | 2.575 | 1 | 0 | 1 | 38 |
| | 1 | 2180 | MSSP | | 1 | 2.569 | 2 | 0 | 2 | 42 |
| | **1** | **294** | **MRSP** | **1** | | **2.614** | **1** | **0** | **7** | **38** |
| Dog 9 | 2 | 2181 | MSSP | | 2 | 2.647 | 1 | 1 | 2 | 41 |

[a]Raw data for each isolate are available in Table S1. Information on the representative isolate for each MLST group is available in Table S4.

as shown by ST2180 and ST294 clustering apart from ST2179 and unknown MLSTs from the same dog.

Four dogs harbored a unique MLST (Dog 1 unknown ST, Dog 2 unknown ST, Dog 7 ST1026, and Dog 9 ST2181), showing a homogeneous MLST distribution, whereas the rest of the dogs showed a heterogeneous distribution (Fig. 2). The dog with the more heterogeneous population was Dog 8, which harbored four different MLSTs, and it was followed by Dog 5 and Dog 6, which harbored three different MLSTs (Fig. 2). We observed that in some dogs, the same MLST was present in different body sites (Dog 1, Dog 6, Dog 7, Dog 8), whereas for other dogs, the same MLST was identified in a particular body site (Dog 3, Dog 5, Dog 9).

Genomic analyses revealed that all of the isolates analyzed herein were assembled into a main complete contig that corresponded to the bacterial chromosome, with a median genome size of 2.60 Mbp (Table S1). By grouping the isolated genomes by MLST and host, we observed variation in the median genome sizes, ranging from 2.471 Mbp (ST unknown) to 2.874 bp (ST551) (Table 1). Genomic data also revealed that all of the isolates that were analyzed in this study contained at least one antibiotic resistance gene (Table S1). ST551, ST2177, and three unknown MLSTs from Dog 1, Dog 5, and Dog 6 harbored between 9 and 10 antimicrobial resistance genes (Table 1). The median number of prophages was high in ST551, ST1026, and ST2177, showing between 3 and 4 prophages (Table 1). A second group of MLSTs, namely, ST1276, ST2178, and ST2180, showed a median of two prophages, whereas the rest of MLSTs showed a median of one prophage (Table 1). The presence of plasmids was observed in ST551, ST2178, and ST2181 isolates. The ST551 isolates displayed the largest genome sizes and the highest numbers of antibiotic resistance genes and prophage content (Table 1).

Moreover, we observed a specific MLST distribution, depending on the methicillin resistance genotype. For instance, within isolates from Dog 6, all of the ST551 isolates corresponded to the MRSP genotype, and all of the ST2178 isolates corresponded to the MSSP genotype.

Within the 67 isolates, we could not define a specific ST per body site. Dogs with higher numbers of bacterial isolates with different MLSTs, such as Dog 6 (*n* = 17), showed ST2178 in inguinal and perianal skin, ST551 in perioral and nasal skin, and unknown MLSTs in nasal skin. Similarly, dogs with higher numbers of isolated samples with a unique MLST, such as Dog 7 (*n* = 11), showed ST1026 in perioral, nasal, and

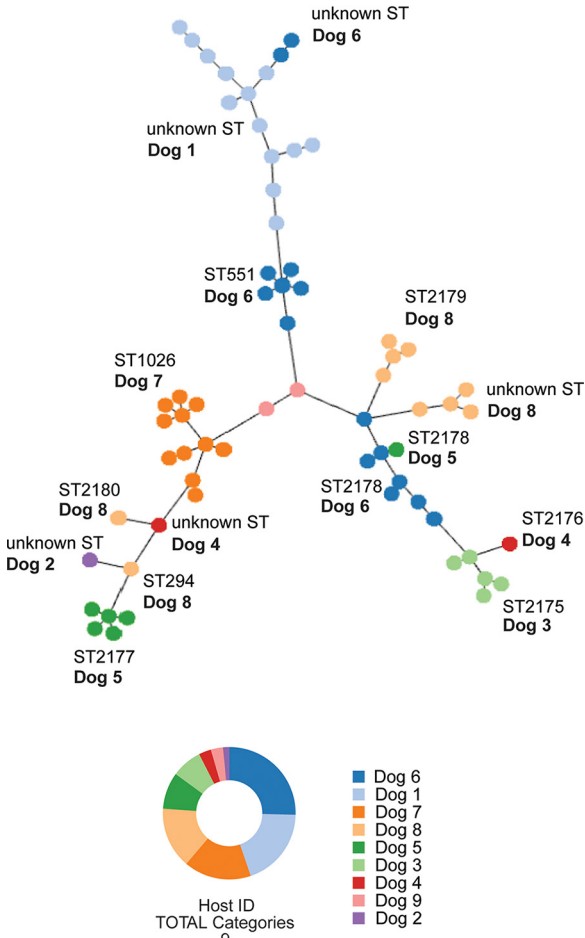

**FIG 1** MLST distribution among hosts. Tree of SNPs, considering the whole-genome sequencing data and the MLST allelic profiles from the 67 *S. pseudintermedius* genomes that were analyzed in this study.

inguinal skin (Fig. 2; Table S3). In conclusion, MLSTs showed a heterogeneous distribution among different body sites.

While we obtained 67 complete genomes of *S. pseudintermedius*, we identified ANI percentage identity values over 99.9% between those isolates corresponding to the same MLST (Table S4). We selected one representative for isolates that belonged to the same MLST and the same dog. Therefore, 18 representative *S. pseudintermedius* genomes were considered for further analyses (Table S5).

We carried out a statistical analysis to compare the genomic data of the 18 *S. pseudintermedius* representative isolates between the four body sites that were analyzed in this study: inguinal skin ($n = 4$), nasal skin ($n = 5$), perianal skin ($n = 6$), and perioral skin ($n = 3$). We did not observe any significant differences between the diverse body sites in terms of the MLST, genome size, or numbers of antibiotic resistance genes, plasmids, prophages, or virulence factors (Fig. S1).

**S. pseudintermedius populations from healthy dogs harbor both MRSP and MSSP genotypes.** Based on the *in silico* detection of the *mecA* gene, 39% of the *S. pseudintermedius* representative genotypes that were carried by healthy dogs were methicillin-resistant. Specifically, 7 out of the 18 representative isolated genomes (39%) were MRSP, and the other 11 isolates (61%) were MSSP (Fig. 3; Table S5). Several differences were detected by comparing the MLSTs with a MRSP genotype against MLSTs with a MSSP genotype.

An exploratory data analysis showed that MRSP isolates generally display larger genomes (range: 2.597 to 2.876 Mbp) than do MSSP isolates (range: 2.471 to 2.646 Mbp),

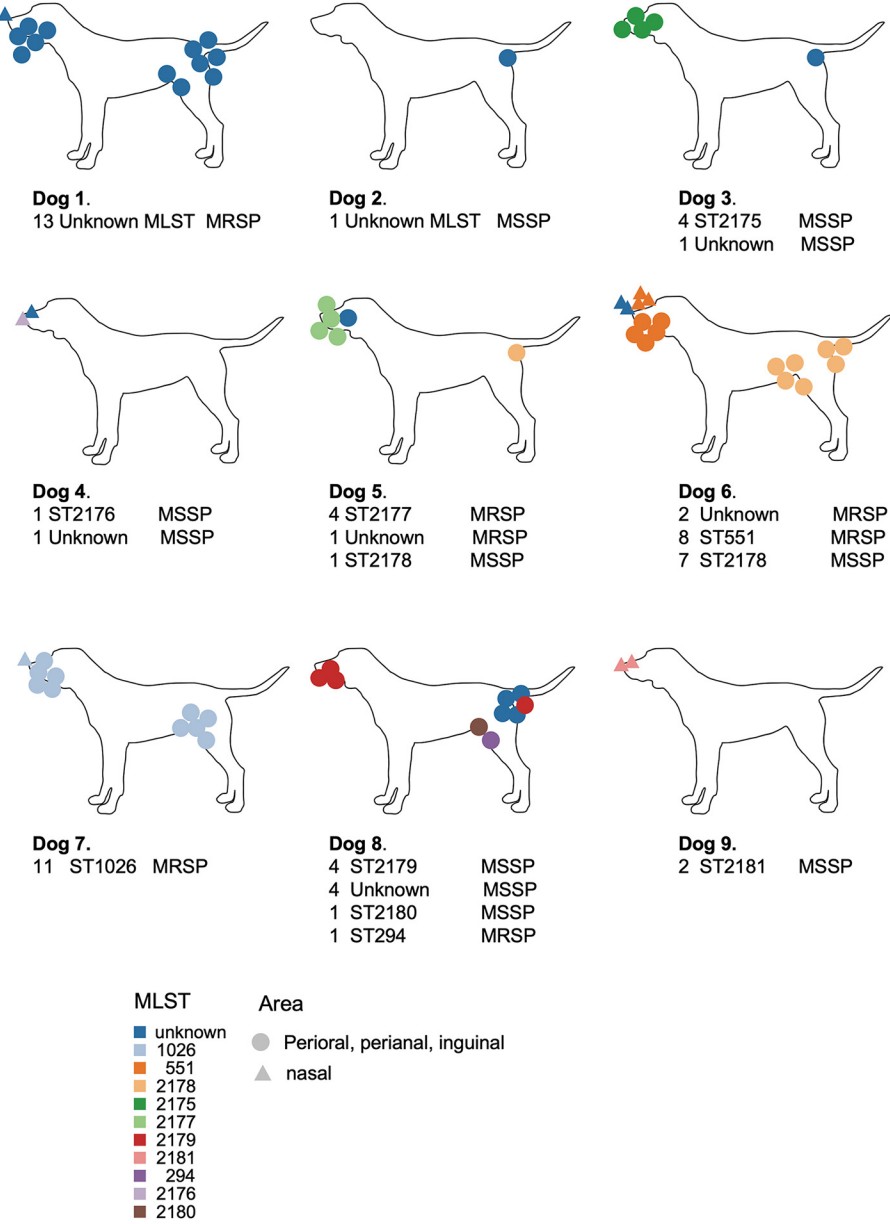

**FIG 2** MLST localization and distribution in each host and body site. A schematic cartoon depicts the different MLSTs that were isolated from the different skin body sites from each healthy dog that was analyzed in this study.

and they also show higher numbers of antibiotic resistance genes (range: 3 to 10), compared to MSSP isolates (range: 1 to 2) (Fig. 3; Table S5). Two dogs showed a homogenous MRSP distribution (Dog 1, Dog 7), four dogs showed a homogenous MSSP distribution (Dog 2, Dog 3, Dog 4, Dog 9), and three dogs showed both MRSP and MSSP genotypes (Dog 5, Dog 6, Dog 8) (Fig. 3).

Supporting the previous exploratory analyses, we carried out statistical analyses on the 18 representative isolates from each dog, confirming that the MRSP genomes were, on average, larger than those from the MSSP isolates (median of 2.646 Mbp versus 2.575 Mbp, respectively; adjusted *P* value = 0.0049) (Fig. 4A). In agreement, the MRSP isolates showed, on average, a significantly higher number of antibiotic resistance genes (median of 9 versus 1; adjusted *P* value = 2E−07) (Fig. 4B), compared to the MSSP isolates. The numbers of prophages, plasmids and virulence factors were the same in the MRSP and MSSP isolates (Fig. 4C–E).

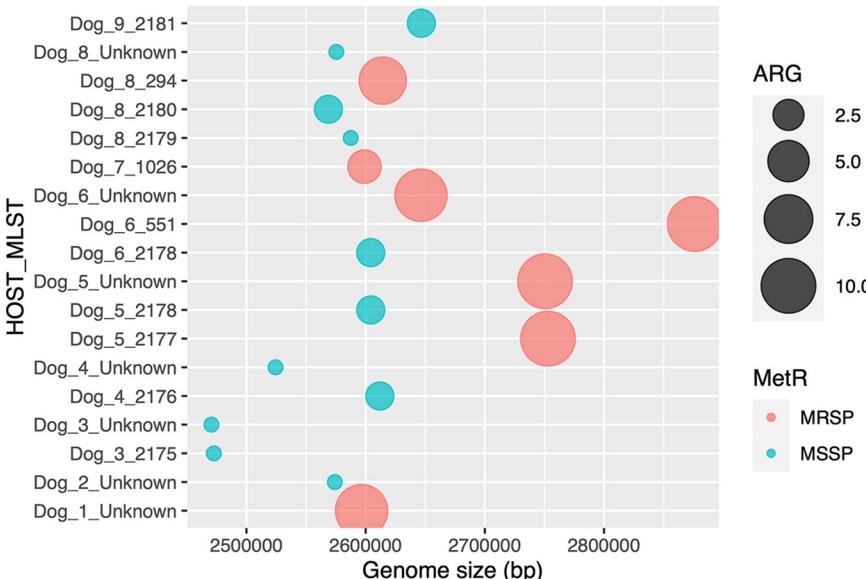

**FIG 3** Representative MLST isolates display different genome sizes and methicillin resistance genotypes. Bubble plots display the genome sizes of the *pseudintermedius* representative isolates, grouped by MLST and host. The size of each bubble indicates the average number of antimicrobial resistance genes. The colors indicate the methicillin resistance genotypes.

Within the 20 *S. pseudintermedius* isolates containing plasmids, 11 were MRSP, and 9 were MSSP, revealing no direct correlation between the methicillin resistance genotype and the plasmid content. Moreover, six MRSP isolates showed associated plasmids with small drug resistance family proteins, and two MSSP isolates contained plasmids with the aminoglycoside 6-nucleotidyltransferase *ANT(6)-I* antibiotic resistance gene (Table S2).

Taken together, these results indicate that the 39% of the *S. pseudintermedius* representative genotypes that were carried by healthy dogs were MRSP, and that the genome size difference between MRSP and MSSP correlated with the higher number of antibiotic resistance genes that were detected in the MRSP compared to the MSSP.

**Pangenome analyses revealed that MSSP isolates were more diverse and heterogeneous than were MRSP isolates.** Global pangenome analyses of the complete genomes of the 67 *S. pseudintermedius* that were isolated from the skin of healthy dogs revealed 59% (2,045 out of 3,463 gene clusters) of core genome and 41% (1,418 out of 3,463 gene clusters) of accessory genome (Fig. 5). All of the isolates corresponding to the same MLST showed an ANI percentage identity above 99.99% (Fig. 5). The ANI phylogenetic tree clustered the samples by methicillin resistance genotype, grouping the MRSP together and apart from the MSSP except for MRSP HSP283, which grouped within the MSSP cluster. A deeper analysis of the ANI phylogenetic tree revealed that the samples were mainly grouped by MLST (Fig. 5).

Pangenome analyses grouped the MRSP genotypes together and apart from the MSSP genotypes (Fig. 5). By splitting the *S. pseudintermedius* pangenome into MSSP-specific (Fig. S2) and MRSP-specific (Fig. S3) pangenomes, we observed that the core genome was similar between the MRSP and MSSP (66% and 67%) (Fig. S2 and S3). However, the specific singleton genome within the MSSP pangenome was larger than that observed in the MRSP pangenome (7% versus 1%) (Fig. S2 and S3). These results showed that the MSSP pangenome harbored more accessory genes that were unique to a single strain, suggesting that the isolates from the MSSP pangenome were more diverse than were the isolates from the MRSP pangenome. Supporting this, four different MLST were identified within the MRSP isolates, whereas at least six different MLST were identified within the MSSP isolates (Table 1; Fig. 1).

To carry out functional enrichment statistical analyses, we considered 25 genomes of *S. pseudintermedius* from the skin of healthy dogs: the 18 representative genomes of

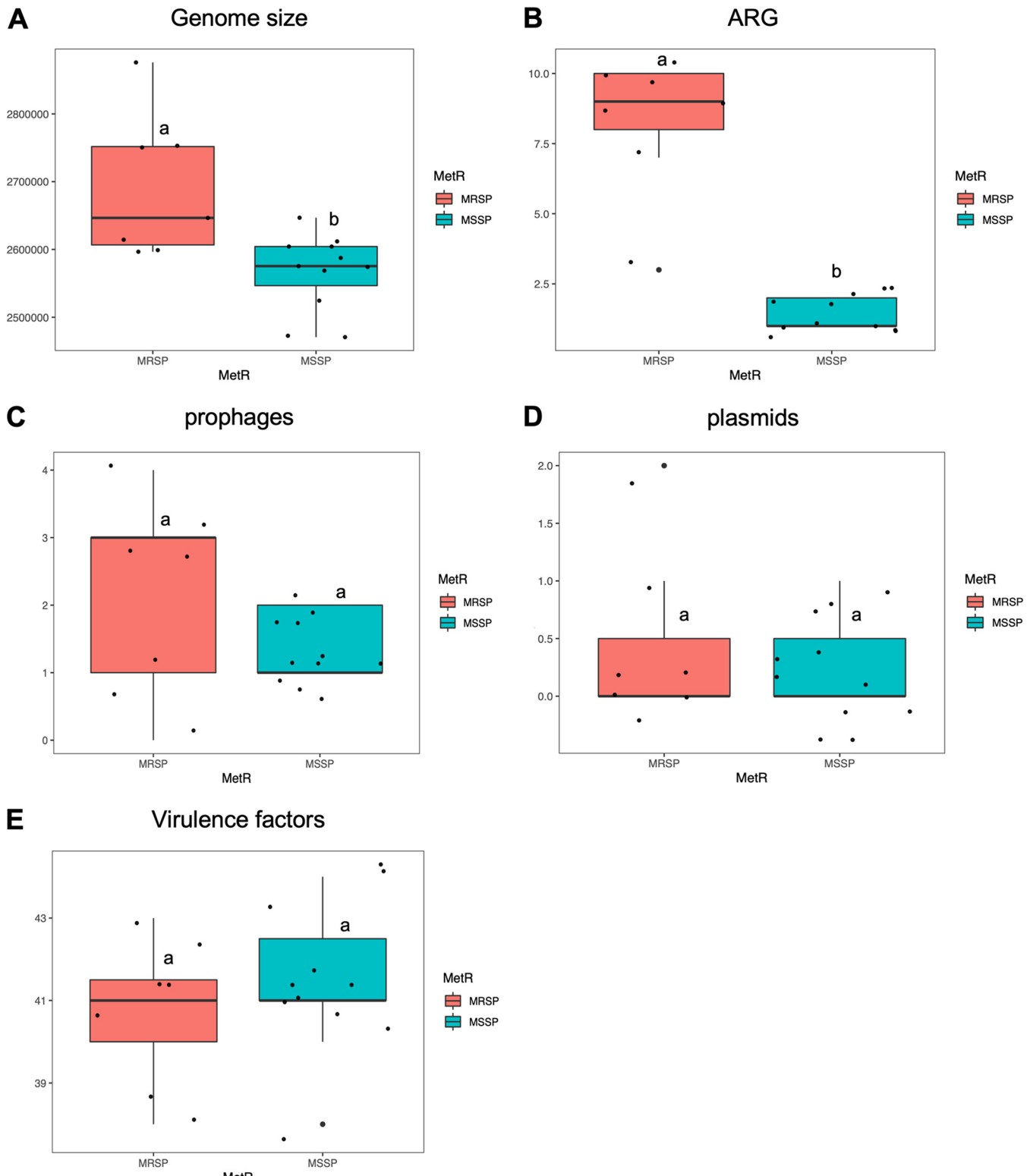

**FIG 4** MRSP genomes are larger and contain more antibiotic resistant genes than MSSP genomes. Box plots show the distribution of the (A) genome size, (B) antimicrobial resistance genes number, (C) prophages number, (D) plasmids number, and (E) virulence factors number of the representative *S. pseudintermedius* isolates that were sequenced in this study. The box plots show the results of the *S. pseudintermedius* genomes (n = 18), comparing the MRSP strains (blue) to the MSSP strains (turquoise). MRSP, n = 7 (39%); MSSP, n = 11 (61%). Shapiro-Wilk normality tests revealed that the data were not normally distributed. A one-way ANOVA with Tukey's test revealed significant differences in those cases, and they are marked with different letters, denoting statistically significant differences (*P* value < 0.05).

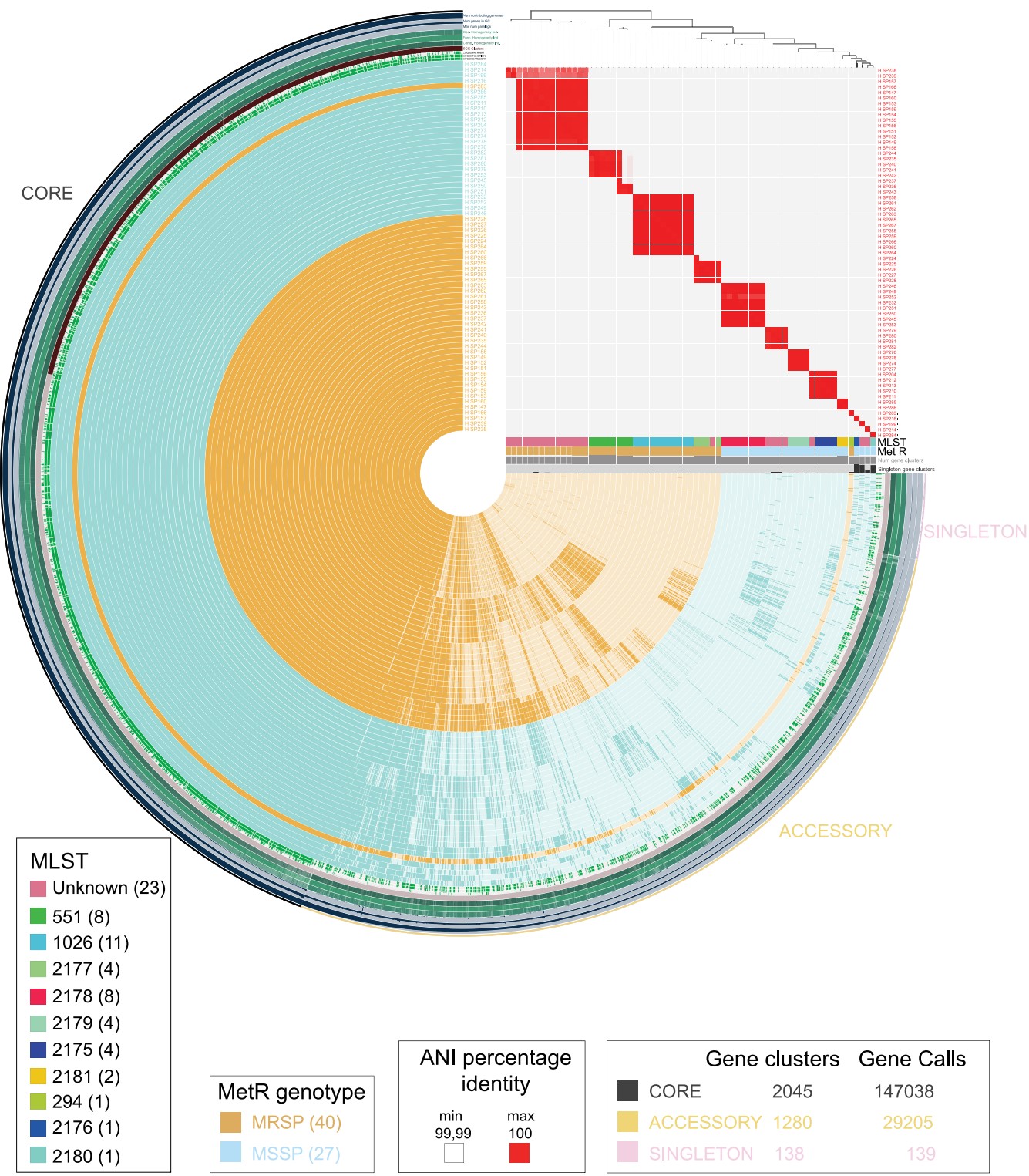

**FIG 5** Pangenome analyses of *S. pseudintermedius* isolated from healthy dogs. Pangenome results of all of the *S. pseudintermedius* genomes (*n* = 67), comparing the MRSP genotypes (blue) to the MSSP genotypes (turquoise). MRSP, *n* = 40 (60%); MSSP, *n* = 27 (40%). By definition, the core genome is the part of the pangenome that is present and shared by all of the genomes within the pangenome. The accessory genome is specific for a group of strains within the pangenome, and singletons are strain-specific genome sequences. The visualization of the pangenome analyses was carried out via ANVI'O. The central dendrogram clustering of samples is ordered by gene cluster presence/absence. Items order: Number of genomes for which each gene cluster has hits (D: undefined; L: undefined). The phylogenetic tree shows the samples, ordered by ANIb percentage identity. Within the phylogenetic tree, each cluster of samples is also represented by a red square, showing the ANI percentage identity values above 99.99% between those isolates corresponding to the same MLST. From left to right, the clusters contain the following samples: Dog 6 unknown MLSTs, Dog 1 unknown MSLTs, Dog 6 ST551, Dog 7 ST1026, Dog 5 ST2177, Dog 6 ST2178, Dog 8 unknown MSLTs, Dog 8 ST2179, Dog 3 ST2175, and Dog 9 ST2181. The last four clusters on the right side contain one single sample each (from left to right): Dog 8 ST294, Dog 2 unknown MLST, Dog 4 ST2176, Dog 4 unknown MLST, and Dog 8 ST2180.

isolates from this study as well as 7 representative genomes of isolates from our previous study (14). Functional enrichment analyses revealed that the MRSP genotypes isolated from healthy dogs were significantly enriched (adjusted $P$ value < 0.05) in six functional categories, compared to the MSSP genotypes: acyl-CoA dehydratase, predicted nucleotidyltransferase, aminoglycoside phosphotransferase (Aph), predicted kinase aminoglycoside phosphotransferase (APT), 3-hydroxy-3-methylglutaryl CoA synthase (PksG), and DNA topoisomerase IA (TopA) (Table S6).

## DISCUSSION

To date, numerous scientific studies have reported genomic data on methicillin-resistant *S. pseudintermedius* (MRSP) pathogenic genotypes that were isolated from dogs with pyoderma. However, WGS studies on commensal *S. pseudintermedius* that inhabit the skin of healthy dogs remain limited. The main objective of this study was to describe the commensal *S. pseudintermedius* populations that inhabit the skin of healthy dogs. We retrieved 67 single main-contig complete *S. pseudintermedius* genomes and 19 associated plasmids that were isolated from the skin of healthy dogs using Nanopore-only long-reads sequencing. The detailed genome information has been previously reported by our group, and the *S. pseudintermedius* genomes were *de novo* assembled with Flye 2.8.3 (12). In the present study, due to the recently improved software, we repeated the *de novo* assembly of the 67 *S. pseudintermedius* genomes with Flye 2.9. The reassembled genomes showed higher completeness, more complete rRNAs, more CDS, and fewer pseudogenes than did previous assemblies, and they were therefore used for further analyses (Table S7).

We reported new allele combinations, resulting in 7 new MLSTs assignations, whereas 3 other MLSTs had already been described. These STs differed from those that were previously isolated from healthy dogs and dogs with pyoderma (13, 14). We previously detected ST71 as the most common ST among the MRSP genotypes that were isolated from dogs with pyoderma (13, 14). In fact, according to the literature, ST71 is the predominant MRSP clone in Europe, and it is followed by the emerging ST551 clone (15). In summary, most of the STs that were identified in our group of healthy dogs were new, and only one ST (ST551 in Dog 6) corresponded to one of the STs that was reported in dogs with pyoderma. This fact seems to contravene the current paradigm, in which pyoderma results from the colonization and subsequent infection by commensal *S. pseudintermedius*. This merits further investigation. Moreover, STs were distributed mainly by dog, rather than by body site. The STs were specific from each dog analyzed within this study (except for one sample). However, we did not observe a specific ST per skin body site. Some STs, such as ST2175 and ST2177, were only observed in perioral skin, but we hypothesize that this could be due to the low number of isolates ($n = 4$).

On average, we found that the MRSP genomes were larger and contained more antibiotic resistance genes than did the MSSP genomes, and this finding is in agreement with the results of our previous work (13, 14). However, the larger average genome sizes that were observed in MRSP isolates were mainly due to the ST551 and ST2177 sequence types. In fact, some MSSP genomes showed similar genome sizes to MRSP genomes. MSSP ST2178 and MRSP ST1026 were approximately 2.60 Mbp. In a related, recently published work (29), we identified that ST551 contained SCC*mec*_subtype-Vc(5C2&5) and that ST2177 contained SCC*mec*_type_IVg(2B) (Table S8). In contrast, unknown MLSTs from dog 1, which were smaller than the MRSP genomes, did not contain any SCC*mec* element (Table S8). These results indicated that SCC*mec* presence is contributing to the larger genome size. However, ST1026 isolates also contained SCC*mec*_type_IVg(2B), and their genome sizes were actually much smaller (Table S8). The estimated sizes of the SCC*mec* elements were between 2.3 Kb and 42 Kb (30), whereas the genome size differences between the isolates were above 200 Kb. Another study reported estimated sizes of SCC*mec* elements of approximately 5.9 Kb and 12.28 Kb (31). Therefore, we concluded that even though SCC*mec* elements were partially contributing to the genome size, they were not the only reason. Moreover, as we reported in (29), when *mecA* is present, it usually co-occurs with the presence of many other resistance genes. The high presence of

antibiotic resistance genes from other antibiotic families in both ST551 and ST2177 but absent in ST1026 also contributed to this. For instance, Tn5405-like elements, which are present in ST551 and ST2177 but absent in ST1026, can harbor several antibiotic resistance genes, such as *ant*(6)-*I* (*aadE*), *sat-4*, *aph*(3')-*III*, as well as other genes, such as *ermB*, *dfrG*, and *cat*, which confer resistance to different antibiotic families and can even constitute a multidrug resistant genotype.

We observed that 7 out of the 18 (39%) representative *S. pseudintermedius* isolates from healthy skin were MRSP and that the other 11 isolates (61%) were MSSP. However, the results of our previous studies showed that 100% of the *S. pseudintermedius* isolates from healthy skin were MSSP (13, 14). This difference might be due to the number of sequenced samples, which increased from 22 to 67. Previous studies aiming at the MLST characterization of *S. pseudintermedius* isolates from healthy dogs concluded that the MRSP frequency in healthy dogs changed from none (32) to 1.6% (33), 2,6% (24), 4.6% (34), and up to 8% (35). The MRSP frequency rate increases to 7.4 to 47.9% when both diseased and healthy dogs are screened (36, 37), and it increases to 74% in dogs with superficial pyoderma (38). Here, we detected 39% of MRSP by screening healthy dogs only. This high rate of methicillin resistance could be explained as a consequence of our veterinary hospital being a reference center, meaning that some of the dogs had previously been treated at other small veterinary clinics. Also, it is possible that the percentage has increased in recent years. In agreement with this high rate, a previous characterization of *Staphylococcus* spp. that were isolated from healthy canine skin concluded that over 50% of the strains were multidrug resistant and produced gelatinase, DNase, and lipase (39). However, a larger study that increases the sample size of dogs tested should be conducted to determine whether this high carriage rate is consistent. In conclusion, MRSP, antibiotic resistance, and biofilm forming *S. pseudintermedius* are commonly detected not only in dogs with pyoderma but also in healthy dogs.

The pangenome is defined as the set of genes that is present in a given species, across all isolates, and it can be subdivided into the accessory genome, which is present in only some of the genomes, and the core genome, which is present in all of the genomes (40). Open pangenomes are larger and have a smaller proportion of core genes, whereas closed pangenomes are smaller in size and have a larger proportion of core genes (41, 42). The large accessory genome suggests that the *S. pseudintermedius* isolates in this study have an open pangenome. This pangenome shows that the MSSP isolates are more diverse and heterogeneous than are the MRSP isolates, as the MSSP pangenome harbored more accessory genes that were unique to a single strain, in agreement with the results of our previous work (13, 14) as well as with those of other previous studies that reported higher genotypic diversity among MSSP than among MRSP (6).

Functional analyses showed enrichment of aminoglycoside phosphotransferase and kinase aminoglycoside phosphotransferase (Table S6), both of which confer resistance to aminoglycoside antibiotics (43–45). These results are consistent with our findings that show higher numbers of antibiotic resistance genes in the MRSP isolates. Significant enrichment in nucleotidyltransferase and DNA topoisomerase functions suggests that DNA modification events are enriched in MRSP isolates, compared to MSSP isolates. Finally, acyl-CoA dehydratase and 3-hydroxy-3-methylglutaryl CoA synthase suggests that cholesterol and fatty acid metabolism-related functions are enriched in MRSP isolates, compared to MSSP isolates. A previous study showed that the *agr* gene acts as a regulator of fatty acid metabolism, biofilm, and *mecA* expression in methicillin-resistant *Staphylococcus aureus* (46). Understanding the lipid synthesis and secretion of antibiotic-resistant bacteria will be important as a complementary approach to the current focus on resistance genes, such as *mecA*, and such an understanding could provide further guidance toward the development of new strategies by which to overcome antibiotic resistance. In future studies, a higher number of isolates should be sequenced and included so as to obtain more significant insights from the functional enrichment analyses. This information will be relevant to understand the physiology and the biochemical

pathways that are used by commensal *S. pseudintermedius* strains to evolve into pathogenic strains.

In 2019, before starting this project, there were a total of 29 complete genomes of *S. pseudintermedius* that were published in NCBI. Currently, in January of 2023, a total of 144 complete genomes of *S. pseudintermedius* have been announced in NCBI, 101 of which (70%) correspond to our group. Moreover, a total of 45 complete genomes have been annotated by NCBI RefSeq, 11 of which (25%) correspond to our group (https://www.ncbi.nlm.nih.gov/data-hub/genome/?taxon=283734).

It should be noted that all of the *S. pseudintermedius* genomes from this study have been sequenced with Nanopore long-reads, which produces more contiguous genomes (26, 27, 47) and allows for the identification of mobile genetic elements and their locations with greater precision than do other sequencing techniques that are based on short-reads (48, 49). All of the genomes analyzed herein were highly contiguous, as they were all assembled in one single main contig. We also identified the associated plasmids in different isolates. Although it was previously necessary to carry hybrid assemblies combining long-read sequencing with short-read sequencing, we obtained high-quality complete genomes via long-read sequencing only, in agreement with recent studies (26, 27). Together with our previous studies, we demonstrated that Nanopore sequencing data allows for the *de novo* assembly of the entire genome of *S. pseudintermedius* (12–14). Until now, we have announced 95 *S. pseudintermedius* complete genomes that were isolated from the skin of healthy dogs and 33 that were isolated from the lesional skin from dogs with pyoderma, and we have characterized them at the genomic and functional levels.

Finally, our results revealed that the *S. pseudintermedius* populations inhabiting the skin of healthy dogs are relatively diverse and heterogeneous. Indeed, in our study, at least 10 different MLSTs were identified within 9 healthy dogs. Although there is some diversity in the combinations of MLSTs that were identified on the skin of dogs, three main patterns have emerged: (i) a single MLST occupies the entire skin surface; (ii) one MLST occupies or predominates a specific anatomical region within the animal; (iii) two MLSTs coinhabit the same anatomical region from the same animal. In dogs in which *S. pseudintermedius* of a single MLST occupy the entire skin surface, it is impossible to exclude that there are other *S. pseudintermedius* with other MLSTs in these animals. The *S. pseudintermedius* populations of healthy dogs are also diverse in terms of methicillin resistance. Two dogs showed a homogenous MRSP distribution, four dogs showed a homogenous MSSP distribution, and three dogs showed both the MRSP and MSSP genotypes. Antibiotic resistance genes were detected in all 67 of the genomes of *S. pseudintermedius* isolates. The percentage of *S. pseudintermedius* isolates that are methicillin-resistant in healthy dogs was high, indicating the extent and severity of the problem of antimicrobial resistance in staphylococci with zoonotic potential. Our study showed that MRSP were detected in healthy dogs with no recent history of antimicrobial therapy, pointing out the need of carrying genomic studies to monitor and treat infections, such as canine pyoderma.

Taken together, our study shows that healthy dogs are colonized with genetically unrelated and diverse *S. pseudintermedius* populations, which highlights the need to sequence multiple isolates from each dog to investigate the pathogenesis of canine pyoderma. In conclusion, we provide valuable genomic and functional enrichment information regarding the methicillin-resistant genotypic groups, which thereby contributes to a better understanding of the pathogenesis of *S. pseudintermedius* infections.

## MATERIALS AND METHODS

**Bacterial cultures.** Samples were obtained from nine healthy adult dogs belonging to different breeds. Before the collection of samples, each dog was clinically examined by a veterinarian to verify that it did not present any skin lesion. Sterile swabs moistened with sterile saline were rubbed for 30 s on four skin anatomical sites: perinasal, perioral, inguinal, and perianal. These four different skin anatomical sites were chosen to represent different types of microbial habitats within the dog: from a region with fur that was mostly dry, such as the groin (inguinal samples), to mucocutaneous areas, such as the muzzle (nasal and perioral samples), and the perianal region, close to the gastrointestinal tract (50, 51).

The swabs were cultured in blood agar at 37°C for 24 h. Colonies that were grown with a clear distinct morphology of *S. pseudintermedius* (small size silver colonies) were seeded/subcultured in 3 mL of BHI at 37°C for 16 h. When possible, up to five colonies from each skin site from each dog were recovered for sequencing. However, in some cases, the growth on the plate was scarce, and the colony morphology was clearly not compatible with Staphylococci.

**DNA extraction and sequencing.** DNA was extracted using a ZymoBIOMICS DNA Miniprep Kit (Zymo Research). DNA quality and quantity were determined using a NanoDrop 2000 Spectrophotometer and a Qubit dsDNA BR Assay Kit (Fisher Scientific). The sequencing libraries were prepared using 200 to 400ng of DNA that were subjected to transposase fragmentation using a Rapid Barcoding Sequencing Kit (SQK-RBK004; Oxford Nanopore Technologies, ONT). Up to 12 barcoded samples were loaded in a MinION FLO-MIN106 v9.4.1 flow cell and sequenced in a MinION Mk1B or Mk1C (ONT). The fast5 files were basecalled and demultiplexed, and the adapters were trimmed using Guppy 5.0.11 (52) (ONT) (–dna_r9.4.1_450bps_sup.cfg) (–config configuration.cfg –barcode_kits SQK-RBK004 –trim_barcodes; min_score threshold default 60). Reads with a quality score of less than 10 were discarded. The run summary statistics were obtained using Nanoplot 1.38.1 (53) (–N50 –fastq).

**Assembly and visualization of the genomes.** Isolates were confirmed as *S. pseudintermedius* via taxonomy assignment, using the EPI2ME What's In My Pot (WIMP) workflow (54). Genomes were *de novo* assembled using Flye 2.9 (55) (–nano-hq). Contigs were polished using Medaka 1.5.0 (56) (medaka_consensus; -m r941_min_sup_g507). Genome completeness and contamination were assessed using CheckM 1.1.3 (lineage_wf) (28). Circlator 1.5.5 was used to rotate the genomes to fix the start position of the contig with the *dnaA* gene (57) (fixstart –min_id 70). When the *dnaA* gene is not identified, Circlator fixes the start position with the gene that is nearest to the middle of the contig, as predicted by Prodigal. Genomes were annotated using the NCBI Prokaryotic Genome Annotation Pipeline (PGAP 5.3) and Prokka 1.14.6 (58, 59), as were the total number of coding sequences, complete rRNA, and tRNA. Multilocus sequence types (MLSTs) were assigned with PubMLST (60) ([https://pubmlst.org], November 2021).

**Multilocus sequence types, antibiotic-resistance genes, virulence factors, and bacteriophages.** For each genome assembly, the allele numbers and MLSTs were determined on the basis of the *S. pseudintermedius* PubMLST database (https://pubmlst.org/organisms/staphylococcus-pseudintermedius) (60) and the MLST 2.0. software and database (https://cge.food.dtu.dk/services/MLST/) (61). CSI Phylogeny 1.4 was used to generate a FASTA file with all of the SNPs aligned for the 67 genomic sequences (https://cge.food.dtu.dk/services/CSIPhylogeny/) (62). The numbers of SNPs that differed between the isolates and the dogs are reflected in Table S9. Both the FASTA file containing the SNPs alignments and the metadata file for the 67 sequences were uploaded to PHYLOViZ (63), which uses the goeBURST algorithm, a refinement of the eBURST algorithm (64), and its expansion to generate a complete minimum spanning tree. Antibiotic resistance genes were identified with Abricate 1.0.1 (65) (https://github.com/tseemann/abricate) with the CARD (66), NCBI, and ARGNNOT (67) databases. Plasmids were identified using PlasmidFinder 2.1 (68). A custom database was also created to analyze the virulence factors (SPVFDB), containing 58 genes encoding virulence factors, including exfoliative toxins, enterotoxins, leukocidins, pore-forming proteins, and intercellular adhesion proteins. Subsequently, the results were filtered by genes with an identity and coverage of ≥95%. Phigaro 2.3.0 (69) and Virsorter 1.0.6 (70) were used to identify bacteriophage sequences within the genomes.

**Plasmid characterization.** Potential plasmid contigs were assembled using Flye 2.9, the most recently updated version that performs a better assembly of short sequences (plasmids or viruses) that were often missed in previous versions. After assembly, the small additional contigs were further characterized via the examination of the Flye 2.9 outputs: contig size, contig coverage, and contig circularity. Plasmid coverage is usually much higher than the main contig coverage. Abricate PlasmidFinder (68). and oriTfinder (71) were used to identify oriT origins of replication and relaxases. Plasmids were annotated using the PATRIC RASTtk-enabled Genome Annotation Service (72), PGAP 5.3 (59), and Prokka 1.14.6 (58). BLASTn NCBI was used to further analyze the candidate plasmid contigs.

**Statistical analyses and visualization.** R and ggplot were used for the statistical analyses and the box plot visualization. The Shapiro-Wilk normality test was used to analyze whether the data were normally distributed in the healthy *S. pseudintermedius* population. When the data distribution, as determined by the Shapiro-Wilk test, was significantly non-normal ($W = 0.93$, $P < 0.01$), a parametric test was used. The Wilcoxon rank sum exact test revealed a statistically significant difference. A one-way analysis of variance (ANOVA) and Tukey's test revealed statistically significant differences in cases marked by different letters, denoting statistically significant differences ($P < 0.05$).

**Pangenome characterization and visualization.** The pangenome analyses were carried out, following the Anvi'o 7 pangenomics workflow (73) (https://merenlab.org/2016/11/08/pangenomics-v2/). Anvi'o7 allows for the comparison of shared genes, and it was used to determine the core genome and accessory genomes of the *S. pseudintermedius* isolates. Within the Anvi'o pangenomics workflow, Prodigal (74) was used as a gene caller to identify open reading frames, whereas genes were functionally annotated using BLASTP against the NCBI COGs database (75). The pangenome database was created using NCBI's BLASTP to calculate the similarity of each amino acid sequence in every genome against every other amino acid sequence across all genomes to resolve gene clusters. The Markov cluster (MCL) inflation parameter was set to 10. pyANI was used to plot and calculate the ANI values between the genomes. Global pangenome analyses were carried out, considering the 67 *S. pseudintermedius* isolates that were sequenced in the present study. Functional enrichment analyses were carried out, considering only the 18 representative genomes of *S. pseudintermedius* from this study, together with 7 representative genomes of *S. pseudintermedius* that were isolated from healthy dogs in our previous studies: H_SP081, H_SP093, H_SP118, H_SP125, H_SP127, H_SP141, and H_SP142 (14).

**Data availability.** The descriptions and accession numbers of the standardized isolates are presented in Table S1. The genome assemblies and genomic data are publicly available in GenBank (BioProject PRJNA685966). The raw data are available from the Sequence Read Archive (SRA) (BioProject PRJNA685966).

## SUPPLEMENTAL MATERIAL

Supplemental material is available online only.

**SUPPLEMENTAL FILE 1**, XLSX file, 0.4 MB.
**SUPPLEMENTAL FILE 2**, PDF file, 11.4 MB.

## ACKNOWLEDGMENTS

The funding sources were as follows: Research Project RTI2018-101991-B-I00 ("From whole-genome sequencing to clinical metagenomics: investigations on the pathogenesis of *Staphylococcus pseudintermedius* pyoderma in the dog"); Spanish Ministry of Science, Innovation and Universities. Torres Quevedo Project, PTQ2018-009961, co-financed by the European Social Fund and the Spanish Ministry of Science and Innovation; Industrial Doctorate Program Grant 2017DI037 AGAUR, Generalitat de Catalunya, Spain.

L.F. has received unrelated honoraria for lecturing from Zoetis, Bayer, LETI, CEVA, and Affinity Petcare. O.F., N.F., and A.C. have received travel bursaries for lecturing from Oxford Nanopore Technologies.

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
