## [Reviewer comments · Microbiology Spectrum]

Microbiology Spectrum

Diverse populations of *Staphylococcus pseudintermedius* colonize the skin of healthy dogs

Norma Fàbregas Vallvé, Daniel Pérez, Joaquim Viñes, Anna Cuscó Martí, Lourdes Migura-García, Lluís Ferrer, and Olga Francino

Corresponding Author(s): Norma Fàbregas Vallvé, Vetgenomics SL

Review Timeline:

Submission Date:	August 26, 2022
Editorial Decision:	December 14, 2022
Revision Received:	January 13, 2023
Editorial Decision:	January 25, 2023
Revision Received:	January 26, 2023
Accepted:	January 26, 2023

Editor: Matheus Costa

Reviewer(s): The reviewers have opted to remain anonymous.

Transaction Report:

DOI: <https://doi.org/10.1128/spectrum.03393-22>

December 14, 2022

Dr. Norma Fàbregas Vallvé
Vetgenomics SL
R+D
Edifici EUREKA, PRUAB, Universitat Autònoma de Barcelona
Bellaterra, Barcelona 08193
Spain

Re: Spectrum03393-22 (**Diverse populations of *Staphylococcus pseudintermedius* colonize the skin of healthy dogs**)

Dear Dr. Norma Fàbregas Vallvé:

Thank you for your patience!

Link Not Available

Sincerely,

Matheus Costa

Journals Department
Reviewer comments:

Reviewer #2 (Comments for the Author):

This is a well-written and well designed study that, through the generation of 67 well-curated complete genomes, will add much to the field of staphylococcus research in veterinary medicine. My main concern is that the sample size of dogs is very small. Although 67 genomes were produced, the number of dogs is still only 9 and I think there needs to be a little more emphasis on the fact that a larger study would be needed to determine if this carriage rate is consistent. 39% is indeed a very high rate of methicillin resistance, could this be the result of the small sample size, isolation technique, other? I would like to see a little more discussion about this and perhaps more of a caveat about stating a 39% resistance rate when the sample size was so small. Still, it is no small feat to curate this number of genomes and the authors should be congratulated for that.

Some comments below:

Some minor grammatical errors in the abstract ie line 19: The 39% of the 30 representative genomes isolated were methicillin-resistant *S. pseudintermedius*

In the 'importance' section it should be noted that *S. pseudintermedius* is a major pathogen of dogs and an occasional zoonotic pathogen.

Line 93: ST71 is also the dominant clone diseased dogs in Oceania (Worthing et al. 2018, Nisa et al 2019)

Line 437: Were the swabs moistened or dry?

Line 169: Were they unknown MLSTs because they contained new alleles not in the database? Please clarify.

Line 259: This sentence is phrased awkwardly- consider '39% of the *S. pseudintermedius* genotypes carried by healthy dogs were methicillin-resistant'

Line 331: One would expect MRSP to have a larger genome because it is known that *mecA* acquisition is via a mobile genetic element. Was the increase in genome size consistent with the average size of SCCmec elements? Even though you received a statistically significant result, you note that some MSSP genomes were large so it is worth noting that a large sample size is needed to conclude whether this observation is true.

Line 344: MRSP carriage rates in healthy dogs have been reported as high as 8% in healthy dogs (Worthing and Brown, 2018)

Line 437: Please describe the source of the animals and selection criteria for what constituted a healthy dog, either by referring to a previous study or writing it here.

Figure 2. It is difficult to see which isolates came from perinasal samples and which from perioral. Could this be depicted more clearly, or in a table (helpful for future studies to know what sample site to compare to)

Figure 5: It is very difficult to discern what the insert with red boxes is showing and the caption didn't help in describing the red component of the figure. The text is very small next to the phylogenetic tree. If the text is unnecessary it should be deleted but if it is meant to be read, it needs to be bigger

Fig 1: Can you include in your results anywhere the number of SNPs that differed between the isolates from within a single dog, and between dogs?

You noted that some dogs carried a single MLST type and some carried several. Do you have data on whether these dogs lived with other dogs or alone, because one might expect to see a more varied microbiome if they have contact with other dogs. If you don't have the data, this might be a point to note in the discussion - that skin microbiome variability can be affected by lifestyle and previous health conditions.

Reviewer #3 (Comments for the Author):

The authors have performed a genomic characterization of *S. pseudintermedius* isolated from various body sites of nine healthy dogs. The only conclusion from this study that speaks to me is that MRSP and MSSP strains can be isolated from healthy dogs and that various genotypes can be isolated from a single animal. My only regret is that the authors did not collect data on the environments in which those animals lived. In my opinion it could have (or may be not) helped to associate environments with genotypes and provide an insight on how *S. pseudintermedius* possibly is transmitted.

However, I found the study to be quite interesting and I do not find experimental design flaws. Although reliance only on assembly from long-reads may not have produced the most accurate assemblies as the authors claim. Nanopore has been known to produce errors too. In this regard they could have run a few hybrid assemblies alongside for comparison's sake. The paper is well written, and the figures and tables are understandable. Therefore, I do not have major concerns. Just a few minor issues for the authors to consider.

1. Line 101 - Please state the hypothesis clearly. This reads like an objective.

2. Line 109 - Do the authors have in mind "PCR-free" protocols? Just in case please check and clarify.

3. Line 144 - The authors collected samples from 4 sites per animal. There were 9 animals in the study, that gives at least 36 isolates. How did they arrive at 67 isolates? Did they isolate multiple isolates from a single sample, and if so, how did they differentiate them?

4. Line 208 - Please place the Table legend at the top of the table.

5. Line 214 - What do the authors mean by "Dogs with the highest number of isolated samples with different MLSTs ..." I thought the number of samples per dog were the same. Or do the authors mean "the number of bacterial isolates"? Please clarify.

6. Line 259 - Okay 39% as it is, is not helpful information. Suppose the authors coupled this with provision of information about the environment in which these dogs live, would it help predict which dogs would likely carry the MRSP?

7. Line 308-310 - Since it appears that the authors have reported these data before and maintain that it is largely in agreement. Perhaps they might as well describe the improvements in the analytical pipeline, that makes these new data more valuable than the previously reported data.

8. Line 360-362 - Could this diversity in the pangenome of MSSP be associated with isolation sites? Perhaps discuss this aspect.

9. Line 438 - "perinasal" Would not this site represent environmental contamination rather than a commensal isolation? Since the dog uses this part of the nose quite frequently to survey its surroundings. Why were the deep nasal swabs collected?

10. Line 439 -441 - It appears the selection of colonies to sequence was not really thorough. Please provide the criterium of selecting the colonies.

11. Line 489 - What is the difference between "phage" and "bacteriophage"? Please cite the sources of definition if there is a difference.

12. Line 521 - Please define what MCL is at first occurrence.

13. Line 552 - the colour legend is not necessary since the figure is labelled with the MLSTs types.

Staff Comments:

Preparing Revision Guidelines

Please return the manuscript within 60 days; if you cannot complete the modification within this time period, please contact me. If you do not wish to modify the manuscript and prefer to submit it to another journal, please notify me of your decision immediately so that the manuscript may be formally withdrawn from consideration by Microbiology Spectrum.

Microbiology spectrum review November 2022

This is a well-written and well designed study that, through the generation of 67 well-curated complete genomes, will add much to the field of staphylococcus research in veterinary medicine. It can be accepted once the comments below are addressed, many of which are minor. My main concern is that the sample size of dogs is very small. Although 67 genomes were produced, the number of dogs is still only 9 and I think there needs to be a little more emphasis on the fact that a larger study would be needed to determine if this carriage rate is consistent. 39% is indeed a very high rate of methicillin resistance, could this be the result of the small sample size, isolation technique, other? I would like to see a little more discussion about this and perhaps more of a caveat about stating a 39% resistance rate when the sample size was so small. Still, it is no small feat to curate this number of genomes and the authors should be congratulated for that.

Some comments below:

Some minor grammatical errors in the abstract ie line 19: The 39% of the 30 representative genomes isolated were methicillin-resistant *S. pseudintermedius*

In the 'importance' section it should be noted that *S. pseudintermedius* is a major pathogen of dogs and an occasional zoonotic pathogen.

Line 93: ST71 is also the dominant clone diseased dogs in Oceania (Worthing et al. 2018, Nisa et al 2019)

Line 437: Were the swabs moistened or dry?

Line 169: Were they unknown MLSTs because they contained new alleles not in the database? Please clarify.

Line 259: This sentence is phrased awkwardly- consider '39% of the *S. pseudintermedius* genotypes carried by healthy dogs were methicillin-resistant'

Line 331: One would expect MRSP to have a larger genome because it is known that *mecA* acquisition is via a mobile genetic element. Was the increase in genome size consistent with the average size of SCCmec elements? Even though you received a statistically significant result, you note that some MSSP genomes were large so it is worth noting that a large sample size is needed to conclude whether this observation is true.

Line 344: MRSP carriage rates in healthy dogs have been reported as high as 8% in healthy dogs (Worthing and Brown, 2018)

Line 437: Please describe the source of the animals and selection criteria for what constituted a healthy dog, either by referring to a previous study or writing it here.

Figure 2. It is difficult to see which isolates came from perinasal samples and which from perioral. Could this be depicted more clearly, or in a table (helpful for future studies to know what sample site to compare to)

Figure 5: It is very difficult to discern what the insert with red boxes is showing and the caption didn't help in describing the red component of the figure. The text is very small next to the phylogenetic tree. If the text is unnecessary it should be deleted but if it is meant to be read, it needs to be bigger

Fig 1: Can you include in your results anywhere the number of SNPs that differed between the isolates from within a single dog, and between dogs?

You noted that some dogs carried a single MLST type and some carried several. Do you have data on whether these dogs lived with other dogs or alone, because one might expect to see a more varied microbiome if they have contact with other dogs. If you don't have the data, this might be a point to note in the discussion – that skin microbiome variability can be affected by lifestyle and previous health conditions.

Response to Reviewers - a Manuscript number Spectrum03393-22

“Diverse populations of *Staphylococcus pseudintermedius* colonize the skin of healthy dogs”

Norma Fàbregas, Daniel Pérez, Joaquim Viñes, Anna Cuscó, Lourdes Migura-García, Lluís Ferrer and Olga Francino

Please find below the point-by-point responses to all the issues raised by the reviewers.

Reviewer comments

Reviewer #2 (Comments for the Author):

This is a well-written and well designed study that, through the generation of 67 well-curated complete genomes, will add much to the field of staphylococcus research in veterinary medicine. My main concern is that the sample size of dogs is very small. Although 67 genomes were produced, the number of dogs is still only 9 and I think there needs to be a little more emphasis on the fact that a larger study would be needed to determine if this carriage rate is consistent. 39% is indeed a very high rate of methicillin resistance, could this be the result of the small sample size, isolation technique, other? I would like to see a little more discussion about this and perhaps more of a caveat about stating a 39% resistance rate when the sample size was so small. Still, it is no small feat to curate this number of genomes and the authors should be congratulated for that.

Our response:

We would like to thank the reviewer for the positive feedback. It is true that the number of dogs in the study was relatively small (9). However, it must be understood that the main aim of the study was not to perform an extensive epidemiological study but to analyse in depth the staphylococcal populations living on the skin of healthy dogs. Thus, from the nine dogs, 67 complete genomes could be isolated, which is a considerable number, allowing some conclusions to be drawn about the diversity of staphylococcal populations in the skin microbiota of a dog. Nevertheless, we agree with the reviewer that even though we provide 67 curated and complete genomes, the sample size of dogs could be increased in order to further confirm our results. A larger study would be very useful, and we have discussed this with more detail in the new version of the manuscript as suggested by the reviewer (**lines 380-388**).

In those lines, we also discussed the high rate of methicillin resistance observed in healthy dogs (39%). It is true that previous studies (including our own publications), have reported a much lower percentage of MRSP when the samples were from the skin of healthy dogs. In the present study, however, we need to indicate that although all dogs were clinically healthy, some frequently visited our veterinary teaching hospital, which is a reference center. Also, it is possible that the percentage has increased in the last years. Finally, many thanks for the congratulations, we have indeed worked as a team to obtain, sequence and curate all these genomes and to extract useful information from them. All the comments and suggestions made by the reviewer have improved the quality and readability of our manuscript and we are very grateful for that.

Some comments below:

Some minor grammatical errors in the abstract ie line 19: The 39% of the 30 representative genomes isolated were methicillin-resistant *S. pseudintermedius*

Our response:

Thanks, we have noted and edited this accordingly (**line 35**), however we stated that they are 18 representative genomes, as they are 18 not 30. We hope that this is what the reviewer meant by correcting the grammatical errors:

“The 39% of the 18 representative genomes isolated herein were methicillin-resistant *S. pseudintermedius* (MRSP)”

In the 'importance' section it should be noted that *S. pseudintermedius* is a major pathogen of dogs and an occasional zoonotic pathogen.

Our response:

Thanks, we have noted and edited this accordingly (**lines 46-48**)

Line 93: ST71 is also the dominant clone diseased dogs in Oceania (Worthing et al. 2018, Nisa et al 2019)

Our response:

Thanks, we have noted and edited this accordingly (**line 103**) and added the two citations:

Combining MALDI-TOF and genomics in the study of methicillin resistant and multidrug resistant *Staphylococcus pseudintermedius* in New Zealand
Nisa et al., Scientific Reports (2019)

Clonal diversity and geographic distribution of methicillin-resistant *Staphylococcus pseudintermedius* from Australian animals: Discovery of novel sequence types
Worthing., Veterinary Microbiology (2018)

Line 437: Were the swabs moistened or dry?

Our response:

The swabs were moistened with sterile saline, we have specified this in material and methods section (**lines 478-480**)

Line 169: Were they unknown MLSTs because they contained new alleles not in the database? Please clarify.

Our response:

Yes, the 7 new MLST were annotated in PubMLST because they were known alleles with new combinations. However, 7 MSLTs were labelled as “unknown” because they contained new alleles that were not in the PubMLST database. We have now specified this in the manuscript (**lines 175-181**)

Line 259: This sentence is phrased awkwardly- consider '39% of the *S. pseudintermedius*

genotypes carried by healthy dogs were methicillin-resistant'

Our response:

Thanks, we agree, and we have edited this accordingly (**lines 244-246**)

Line 331: One would expect MRSP to have a larger genome because it is known that *mecA* acquisition is via a mobile genetic element. Was the increase in genome size consistent with the average size of SCC*mec* elements? Even though you received a statistically significant result, you note that some MSSP genomes were large so it is worth noting that a large sample size is needed to conclude whether this observation is true.

Our response:

We previously checked for the SCC*mec* elements in a related article that we recently published where we analyzed the genotype/phenotype correlation of these isolates:

Concordance between Antimicrobial Resistance Phenotype and Genotype of Staphylococcus pseudintermedius from Healthy Dogs

Joaquim Viñes, Norma Fàbregas, Daniel Pérez, Anna Cuscó, Rocío Fonticoba, Olga Francino, Lluís Ferrer, Lourdes Migura-Garcia
Antibiotics (Basel). 2022 Nov 15;11(11):1625.
doi: 10.3390/antibiotics11111625
<https://www.mdpi.com/2079-6382/11/11/1625>

In this recently published article, specifically in the Supplementary Table 4, we described all the SCC*mec* elements identified within the same 67 isolated *S. pseudintermedius* genomes.

Please see new Supplementary Table 8 (**Table S8**) in our revisited manuscript, which show the SCC*mec* elements detected in each isolated genome. ST551 and ST177 genomes, which are the largest among the MRSP, both contain SCC*mec* elements. Specifically, ST551 contains SCC*mec*_subtype-Vc(5C2&5) and ST2177 contains SCC*mec*_type_IVg(2B). Contrary, unknown MLSTs from dog 1, which are smaller MRSP genomes, do not contain any SCC*mec* element. Thus, we could conclude that SCC*mec* presence is contributing to the larger genome size.

However, ST1026 isolates also contain SCC*mec*_type_IVg(2B) and their genome size is actually much smaller. The estimated size of SCC*mec* elements is between 2.3 Kb and 42 Kb (McClure et al., 2020) while the genome size differences between the isolates are above 200 Kb. Supporting this, another studies reported estimated sizes of SCC*mec* elements around 5.9 Kb and 12.28 Kb (Perreten et al., 2013).

Therefore, we conclude that even though SCC*mec* elements are partially contributing to the genome size, they are not the only reason. The high presence of antibiotic resistance genes from other antibiotic families in both ST551 and ST177 are also contributing to this. In fact, as we reported in Viñes et al 2022, when *mecA* is present it usually correlates/co-occurs with the presence of many other resistance genes.

As for large MSSP genomes, such as ST2181, both genomes contain mobile genetic elements of Aminoglycosides resistant genes encoded in plasmids (4,3Kb) that can also ONLY partially explain their larger genome size compared to other MSSP (please see Supplementary table 2). We agree with the reviewer that a large sample size would help to conclude whether this observation is true.

We have now included this in discussion (**lines 344-368**) including new citations:

A Novel Assay for Detection of Methicillin-Resistant Staphylococcus aureus Directly From Clinical Samples

Jo-Ann McClure, John M. Conly, Osahon Obasuyi, Linda Ward, Alejandra Ugarte-Torres, Thomas Louie and Kunyan Zhang

Front. Microbiol., 18 June 2020

Sec. Antimicrobials, Resistance and Chemotherapy

<https://doi.org/10.3389/fmicb.2020.01295>

Novel Pseudo-Staphylococcal Cassette Chromosome mec Element (SCCmec57395) in Methicillin-Resistant Staphylococcus pseudintermedius CC45

Vincent Perreten, Pattrarat Chanchaithong, Nuvée Prapasarakul, Alexandra Rossano, Shlomo E. Blum, Daniel Elad, Sybille Schwendener

Antimicrob Agents Chemother. 2013 Nov; 57(11): 5509–5515.

<https://doi.org/10.1128/AAC.00738-13>

Line 344: MRSP carriage rates in healthy dogs have been reported as high as 8% in healthy dogs (Worthing and Brown, 2018)

Our response:

Thanks, we have noted and edited this accordingly (**line 377**)

Line 437: Please describe the source of the animals and selection criteria for what constituted a healthy dog, either by referring to a previous study or writing it here.

Our response:

In this revised version, we describe the source of the animals and selection criteria for what constituted a healthy dog (**lines 476-478**):

“Samples were obtained from nine healthy adult dogs belonging to different breeds. Before collecting samples, each dog was clinically examined by a veterinarian to verify that he did not present any skin lesion.”

Figure 2. It is difficult to see which isolates came from perinasal samples and which from perioral. Could this be depicted more clearly, or in a table (helpful for future studies to know what sample site to compare to)

Our response:

Yes, we completely agree on this. We have edited Figure 2 to improve clarity and we have added a figure legend to differentiate the nasal samples (depicted with a triangle) from the perioral samples (depicted with a circle). Moreover, we also provide this specific information for each isolate genome in Supplementary Table 3

Figure 5: It is very difficult to discern what the insert with red boxes is showing and the caption didn't help in describing the red component of the figure. The text is very small next to the phylogenetic tree. If the text is unnecessary it should be deleted but if it is meant to be read, it needs to be bigger

Our response:

Thanks, you are right. However, due to the complexity of Figure 5, it is difficult to make the text next to the phylogenetic tree bigger than it is now. We hope that the final version of the

figure with a higher resolution will be more readable. We have removed the caption next to the phylogenetic tree as it was redundant with the color legend of the MLSTs. Moreover, we have edited the Figure 5 caption including a clear description of the phylogenetic tree clusters, which we believe that will help to interpret the results (lines 638-646)

Fig 1: Can you include in your results anywhere the number of SNPs that differed between the isolates from within a single dog, and between dogs?

Our response:

Yes, we have now included the number of SNPs that differed between the isolates and dogs in Supplementary table 9 – Table S9 (line 526-527)

You noted that some dogs carried a single MLST type and some carried several. Do you have data on whether these dogs lived with other dogs or alone, because one might expect to see a more varied microbiome if they have contact with other dogs. If you don't have the data, this might be a point to note in the discussion - that skin microbiome variability can be affected by lifestyle and previous health conditions.

Our response:

All nine dogs were privately owned pets, living in family houses, most indoor. Some were the single pet in the house, while in some other cases they lived with another dog or a cat. In all cases, however, they were frequently in contact with other dogs. They socialized in parks and, and they frequently visited the Veterinary School or the Veterinary teaching hospital. Because all of them had a very similar lifestyle, we have not correlated this point with the results of the skin microbiome variability.

Reviewer #3 (Comments for the Author):

The authors have performed a genomic characterization of *S. pseudintermedius* isolated from various body sites of nine healthy dogs. The only conclusion from this study that speaks to me is that MRSP and MSSP strains can be isolated from healthy dogs and that various genotypes can be isolated from a single animal. My only regret is that the authors did not collect data on the environments in which those animals lived. In my opinion it could have (or may be not) helped to associate environments with genotypes and provide an insight on how *S. pseudintermedius* possibly is transmitted.

However, I found the study to be quite interesting and I do not find experimental design flaws. Although reliance only on assembly from long-reads may not have produced the most accurate assemblies as the authors claim. Nanopore has been known to produce errors too. In this regard they could have run a few hybrid assemblies alongside for comparison's sake. The paper is well written, and the figures and tables are understandable. Therefore, I do not have major concerns. Just a few minor issues for the authors to consider.

Our response:

We would like to thank the reviewer for the positive feedback. Indeed, data on the environments in which the animals lived which, could provide insights on how *S. pseudintermedius* is transmitted. Unfortunately, we do not have detailed information of the environment in which these dogs lived. However, all nine dogs were privately owned pets, living in family houses, most indoor. Some were the single pet in the house, while in some other cases they lived with another dog or a cat. In all cases, however, they were frequently in contact with other dogs. They socialized in parks and they frequently visited the Veterinary

School or the Veterinary teaching hospital. Because all of them had a very similar lifestyle, we have not correlated this point with the possible transmission.

Regarding the Nanopore-only assemblies, we agree that hybrid assemblies would be more accurate, yet more expensive, and time consuming. One of the objectives of this work was also to provide a fast and affordable sequencing method and bioinformatic analysis pipeline that can be applied at the veterinary hospitals for clinical diagnostics. Finally, we would like to thank the reviewer for all the insightful comments and suggestions that have improved the quality of our manuscript.

1. Line 101 - Please state the hypothesis clearly. This reads like an objective.

Our response:

Thanks, we have noted and edited this accordingly (lines 111-114)

2. Line 109 - Do the authors have in mind "PCR-free" protocols? Just in case please check and clarify.

Our response:

Yes you are right, we have edited this accordingly (line 119)

3. Line 144 - The authors collected samples from 4 sites per animal. There were 9 animals in the study, that gives at least 36 isolates. How did they arrive at 67 isolates? Did they isolate multiple isolates from a single sample, and if so, how did they differentiate them?

Our response:

We isolated up to 5 independent colonies (A-E) from each animal body site (4) for the 9 dogs. This information is reflected in Supplementary table 1 (Dog, Area, Colony). In some cases growth of bacteria on the plate was scarce or colonies were not compatible with *S. pseudintermedius* morphology. We sequenced all colonies morphologically compatible with *S. pseudintermedius* (small silver colonies) and some colonies that we have doubts about. Some of the sequenced colonies ended up not being *S. pseudintermedius*, and 67 were confirmed to be *S. pseudintermedius* by Nanopore sequencing. We have described this further in material and methods (lines 484-489)

4. Line 208 - Please place the Table legend at the top of the table.

Our response:

Thanks, we have edited this accordingly (lines 218-220)

5. Line 214 - What do the authors mean by "Dogs with the highest number of isolated samples with different MLSTs ..." I thought the number of samples per dog were the same. Or do the authors mean "the number of bacterial isolates"? Please clarify.

Our response:

Yes, the number of samples collected per dog were the same. However, not all the samples corresponded to *Staphylococcus pseudintermedius*. We meant the number of bacterial isolates, thanks for noticing, we have edited this (line 224)

6. Line 259 - Okay 39% as it is, is not helpful information. Suppose the authors coupled this with provision of information about the environment in which these dogs live, would it help predict which dogs would likely carry the MRSP?

Our response:

Unfortunately, we do not have detailed information of the environment in which these dogs lived. However, we know that they were family-owned pets, that were in frequent contact with other pets. Unfortunately, as you indicate, from our data it is not possible to predict which dogs would likely carry MRSP

7. Line 308-310 - Since it appears that the authors have reported these data before and maintain that it is largely in agreement. Perhaps they might as well describe the improvements in the analytical pipeline, that makes these new data more valuable than the previously reported data.

Our response:

First of all, we wanted to make sure that the reviewer refers to the following sentence:

“To carry out functional enrichment statistical analyses, we considered 25 genomes of *S. pseudintermedius* from the skin of healthy dogs: the 18 representative genomes of isolates from this study, and 7 representative genomes of isolates from our previous study (14).”

In this particular sentence, we refer to a previous study that we published a year ago:

Ferrer L, García-Fonticoba R, Pérez D, Viñes J, Fàbregas N, Madroñero S, Meroni G, Martino PA, Martínez S, Maté ML, Sánchez-Bruni S, Cuscó A, Migura-García L, Francino O. 2021. Whole genome sequencing and de novo assembly of *Staphylococcus pseudintermedius*: a pangenome approach to unravelling pathogenesis of canine pyoderma. *Vet Dermatol* 32.

In that study, we basically used the same analytical pipeline (please see material and methods). However, the goal of that study was to compare *Staphylococcus pseudintermedius* isolated from dogs with pyoderma vs *Staphylococcus pseudintermedius* isolated from healthy dogs.

In the present study, we aimed to characterize the genomic (landscape and) variability of *Staphylococcus pseudintermedius* isolated from the skin of healthy dogs. Therefore, in order to increase the population of *Staphylococcus pseudintermedius* genomes isolated from healthy dogs in our functional analyses, we included 7 representative genomes isolated from healthy dogs that were sequenced in our previous study (Ferrer et al 2021).

This is also described in material and methods section, specifically in **lines 572-577**:

“Global pangenome analyses were carried considering the 67 *S. pseudintermedius* isolates sequenced in the present study. Functional enrichment analyses were carried considering only the 18 representative genomes of *S. pseudintermedius* from this study together with 7 representative genomes of *S. pseudintermedius* isolated from healthy dogs in our previous studies: H_SP081, H_SP093, H_SP118, H_SP125, H_SP127, H_SP141, H_SP142 (14).”

In the other hand, we have previously reported these data before in a short genome announcement paper (MRA):

Fàbregas N, Pérez D, Viñes J, Fonticoba R, Cuscó A, Migura-García L, Ferrer L, Francino O. 2022. Whole-Genome Sequencing and De Novo Assembly of 67 *Staphylococcus pseudintermedius* Strains Isolated from the Skin of Healthy Dogs . *Microbiol Resour Announc* 11.

We used the same exact pipeline as described in material and methods. However, Nanopore recently improved the analysis software (Flye 2.8 was updated to version Flye 2.9) that currently allows nearly finished bacterial genomes without short-read polishing (Sereika *et al.*, 2021; Sereika *et al.*, 2022) and a better assembly of the associated plasmids and virus sequences (<https://github.com/fenderglass/Flye>).

We describe the improvement of Flye 2.8.3 vs Flye 2.9 in Table S7 and in **lines 320-326**:

“The detailed genome information has been previously reported by our group yet the *S. pseudintermedius* genomes were *de novo* assembled with Flye 2.8.3 (12). In the present study, due to the recently improved software, we repeated the *de novo* assembly of the 67 *S. pseudintermedius* genomes with Flye 2.9. The re-assembled genomes showed higher completeness, more complete rRNAs, more CDS, and fewer pseudogenes than previous assemblies and therefore they were used for further analyses (Table S7).”

8. Line 360-362 - Could this diversity in the pangenome of MSSP be associated with isolation sites? Perhaps discuss this aspect.

Our response:

The MSSP from this study were isolated from the same isolation sites than the MRSP. When we compare MSSP vs MRSP isolates we observe different genome sizes and different number of antibiotic resistant genes (Figure 3). However, when we compared isolates by body skin site or skin area, we did not observe any statistical difference in genome size nor antibiotic resistant genes number (Supplementary Figure 1). This observation suggests that probably neither MSSP nor MRSP are associated to a specific isolation site in our study. Also, by looking into Supplementary table 3, we can confirm this by analyzing each body site: inguinal (5 MSSP, 8 MRSP), nasal (4 MSSP, 7 MRSP), perianal (11 MSSP, 5MRSP), perioral (7 MSSP, 20 MRSP).

9. Line 438 - "perinasal" Would not this site represent environmental contamination rather than a commensal isolation? Since the dog uses this part of the nose quite frequently to survey its surroundings. Why were the deep nasal swabs collected?

Our response:

We did not collect deep nasal swabs but perinasal swabs. In our experience, the perinasal samples are representative of the commensal bacteria, specifically, a representative population from the mucocutaneous areas. Often, we identify the same MLSTs in perinasal and perioral samples. For example, in Dog 6, Nasal and Perioral samples are both ST551, or in Dog 5, Nasal and Perioral samples are both ST1026 please see Supplementary Table 3). However, we cannot rule out that we are also identifying environmental contamination.

We have described this further in material and methods section (**lines 476-483**):

“Samples were obtained from nine healthy adult dogs belonging to different breeds. Before collecting samples, each dog was clinically examined by a veterinarian to verify that he did not present any skin lesion. Sterile swabs moistened with sterile saline were rubbed for 30 seconds on four skin anatomical sites: perinasal, perioral, inguinal, and perianal. These four different skin anatomical sites were chosen to represent different types of microbial habitat within the dog: from a region with fur and mostly dry like the groin (inguinal samples) to mucocutaneous areas like the muzzle (nasal and perioral samples), and the perianal region, close to the gastrointestinal tract (50, 51).”

10. Line 439 -441 - It appears the selection of colonies to sequence was not really thorough. Please provide the criterium of selecting the colonies.

Our response:

In this revised version of the manuscript, we have further described the collection of swab samples and the selection of the colonies in material and methods section (**lines 484-489**):

“The swabs were cultured in blood agar at 37°C for 24 hours. Colonies grown with clear distinct morphology of *S. pseudintermedius* (small size silver colonies) were seeded/sub-cultured in 3ml of BHI at 37°C for 16 hours. When possible, up to five colonies from each skin site from each dog were recovered for sequencing. However, in some cases, the growth on the plate was scarce and colony morphology was clearly not compatible with *Staphylococci*.”

11. Line 489 - What is the difference between "phage" and "bacteriophage"? Please cite the sources of definition if there is a difference.”

Our response:

Thanks for noticing this error, there are no differences between phage and bacteriophage. We have corrected this in the new revisited version of the manuscript (**line 538**)

12. Line 521 - Please define what MCL is at first occurrence.

Our response:

MCL is a general purpose cluster algorithm that can be used to extract clusters from networks (https://link.springer.com/protocol/10.1007/978-1-61779-361-5_15)

MCL stands for *Markov Cluster*. The MCL algorithm is a **cluster** algorithm that is basically a shell in which an algebraic process is computed. This process iteratively generates stochastic matrices, also known as **Markov** matrices, named after the famous Russian mathematician Andrei Markov.

We have introduced this in material and methods section (**line 570-571**)

13. Line 552 - the colour legend is not necessary since the figure is labelled with the MLSTs types.

Our response:

In this revised version, we have removed the figure labelling of the Dogs_MLSTs next to the ANI phylogenetic tree as the other reviewer claimed that it was too difficult to read. We have now included this information in the figure 5 legend (**lines 638-646**) and thus, we have kept the colour legend within the figure indicating the different MLSTs

“Phylogenetic tree shows samples ordered by ANIb percentage identity. Within the phylogenetic tree, each cluster of samples is also represented by a red square, showing ANI percentage identity values above 99,99% between those isolates corresponding to the same MLST. From left to right the clusters contain the following samples: Dog 6 unknown MLSTs, Dog 1 unknown MSLTs, Dog 6 ST551, Dog 7 ST1026, Dog 5 ST2177, Dog 6 ST2178, Dog 8 unknown MSLTs, Dog 8 ST2179, Dog 3 ST2175, Dog 9 ST2181. Last four clusters on the right side contain one single sample each one (from left to right): Dog 8 ST294, Dog 2 unknown MLST, Dog 4 ST2176, Dog 4 unknown MLST, Dog 8 ST2180.”

Staff Comments:

Preparing Revision Guidelines

- Point-by-point responses to the issues raised by the reviewers in a file named "Response to Reviewers," NOT IN YOUR COVER LETTER.

INCLUDED

- Upload a compare copy of the manuscript (without figures) as a "Marked-Up Manuscript" file.

INCLUDED

- Each figure must be uploaded as a separate file, and any multipanel figures must be assembled into one file.

INCLUDED

- Manuscript: A .DOC version of the revised manuscript

INCLUDED

- Figures: Editable, high-resolution, individual figure files are required at revision, TIFF or EPS files are preferred

INCLUDED

Please return the manuscript within 60 days; if you cannot complete the modification within this time period, please contact me. If you do not wish to modify the manuscript and prefer to submit it to another journal, please notify me of your decision immediately so that the manuscript may be formally withdrawn from consideration by Microbiology Spectrum.

January 25, 2023

Dr. Norma Fàbregas Vallvé
Vetgenomics SL
R+D
Edifici EUREKA, PRUAB, Universitat Autònoma de Barcelona
Bellaterra, Barcelona 08193
Spain

Re: Spectrum03393-22R1 (**Diverse populations of *Staphylococcus pseudintermedius* colonize the skin of healthy dogs**)

Dear Dr. Norma Fàbregas Vallvé:

Dear Authors,

Please see below very minor modifications suggested by one of the reviewers. I trust you will find them straightforward.

Link Not Available

Sincerely,

Matheus Costa

Journals Department
Reviewer comments:

Reviewer #2 (Comments for the Author):

Thank you for your revisions. I have 2 minor suggestions:

In the abstract, this sentence needs to have the word 'The' removed from the start: 'The 39% of the 18 representative genomes isolated herein were methicillin-resistant *S. pseudintermedius* (MRSP) and showed, on average, a higher number of antibiotic resistance genes and prophages than the methicillin-sensitive (MSSP).'

I think that Figure 1 would be more informative if the dots were colored by dog, not MLST. The more interesting finding for me is how related the isolates were within and between dogs, not the name of the MLST on each dog.

Reviewer #3 (Comments for the Author):

Thank you for addressing the all my comments and suggestions.

Staff Comments:

Preparing Revision Guidelines

Please return the manuscript within 60 days; if you cannot complete the modification within this time period, please contact me. If you do not wish to modify the manuscript and prefer to submit it to another journal, please notify me of your decision immediately so that the manuscript may be formally withdrawn from consideration by Microbiology Spectrum.

Response to Reviewers - a Manuscript number Spectrum03393-22

“Diverse populations of *Staphylococcus pseudintermedius* colonize the skin of healthy dogs”

Norma Fàbregas, Daniel Pérez, Joaquim Viñes, Anna Cuscó, Lourdes Migura-García, Lluís Ferrer and Olga Francino

Please find below the point-by-point responses to the minor revisions raised by the reviewers.

Reviewer comments:

Reviewer #2 (Comments for the Author):

Thank you for your revisions. I have 2 minor suggestions:

In the abstract, this sentence needs to have the word 'The' removed from the start: 'The 39% of the 18 representative genomes isolated herein were methicillin-resistant *S. pseudintermedius* (MRSP) and showed, on average, a higher number of antibiotic resistance genes and prophages than the methicillin-sensitive (MSSP).'

Our response:

Ok, thanks, we have removed “The” from the sentence (line 35).

I think that Figure 1 would be more informative if the dots were colored by dog, not MLST. The more interesting finding for me is how related the isolates were within and between dogs, not the name of the MLST on each dog.

Our response:

We have modified Figure 1 and now the SNP tree is showing the dots colored by dog instead of MLST. We would like to thank the reviewer for the insightful comments and suggestions that have improved the quality and readability of our manuscript.

Reviewer #3 (Comments for the Author):

Thank you for addressing the all my comments and suggestions.

Our response:

Thanks to the reviewer for the insightful comments and suggestions that have improved the quality and readability of our manuscript.

January 26, 2023

Dr. Norma Fàbregas Vallvé
Vetgenomics SL
R+D
Edifici EUREKA, PRUAB, Universitat Autònoma de Barcelona
Bellaterra, Barcelona 08193
Spain

Re: Spectrum03393-22R2 (**Diverse populations of *Staphylococcus pseudintermedius* colonize the skin of healthy dogs**)

Dear Dr. Norma Fàbregas Vallvé:

Your manuscript has been accepted, and I am forwarding it to the ASM Journals Department for publication. You will be notified when your proofs are ready to be viewed.

Sincerely,

Matheus Costa
Editor, Microbiology Spectrum
